# The temporal distribution of ridership in metro stations from land-use perspective

Shian Dai[1]*, Liqiang Yu[2], Lang Song[1], Ying Li[1], Xuze Fan[1]

**1** School of Civil and Architectural Engineering, Xi'an Jiaotong University City College, Xi'an, Shaanxi Province, China, **2** China Northwest Architectural Design and Research Institute CO., LTD, Xi'an, Shaanxi Province, China

* dsa1019@foxmail.com

**Data Availability Statement:** All relevant data are within the manuscript and its Supporting Information files.

**Funding:** This study is jointly supported by the Shaanxi Provincial Department of Science and

## Abstract

A reasonable land use development around subway stations can balance the utilization rates of the subway system during peak and off-peak hours, thereby enhancing its service levels and operational efficiency. Analyzing the temporal distribution patterns of passenger flow and their influencing factors is crucial for determining the optimum ratio of each land use type surrounding metro stations. Thus, this paper employs principal component analysis (PCA) at first to investigate the temporal distribution of metro ridership, and identify their main patterns and factor loadings. Then, using geographically weighted regression, the study examines the spatial dependencies between the main component proportions and influencing factors, focusing on Xi'an subway stations. The results indicate that the temporal distribution of passenger flow can be decomposed into three principal components: the first representing commuting characteristics, and the second and third representing regulating functions. The overall distribution is a composite of these components in varying proportions. Residential and educational land uses primarily drive morning and evening peak flows, with residential land use in the city center and peripheral areas having a more pronounced effect compared to transitional areas. Conversely, commercial & office, healthcare, and recreational & park land mitigate peak flows and increase off-peak flows. External hub enhances passenger flow throughout the day, while industrial land use has negligible impact.

## Introduction

Urban rail transit systems are intricately linked to the land use characteristics of their surroundings, which dictate the origins and destinations of travel and thus affect passenger flows through a phenomenon known as "generation and attraction." By optimizing the mix of land uses along metro corridors, it is possible to effectively balance the usage of metro systems between peak and off-peak periods [1], which allows transit agencies to make better use of existing infrastructure and reduce the need for costly expansions. Additionally, it helps to ensure a more pleasant and reliable service across all times of the day for transit users.

Stations serve as unique nodes within the urban rail transit system, forming the foundational units for route-level and network-wide passenger flows. These flows are directly influenced by surrounding land use, which induces different travel purposes and subsequently impacts the temporal distribution of passengers at each station.

Technology Talent Program [grant number: 2024ZC-KJXX-011], Youth Projects of Xi'an Jiaotong University City College [grant number: 2022Q01], and the Research and Innovation Team of Xi'an Jiaotong University City College: Green Ecological Empowerment and Urban Resilience Innovation Team [grant number: 037010]. All funds were received by Shian Dai. The funders had no role in study design, data collection and analysis, decision to publish, or preparation of the manuscript.

**Competing interests:** The authors have declared that no competing interests exist.

Previous studies have typically examined passenger counts at various temporal granulations, such as daily, weekday and monthly, focusing on boardings and alightings. These studies often rely on static time points rather than time series [2, 3], despite the evident non-stationarity in passenger flows over time. Such approaches fail to capture detailed temporal variations in passenger flows, which hinders the assessment of whether policies effectively optimize flows during targeted time periods [4, 5]. Research has shown that the determinants of peak-period passengers differ from those of off-peak and daily passengers [6, 7]. The impact of various land uses on peak flows differs [8].

By summarizing the full-day time distribution characteristics of each station [9, 10] and conducting cluster analyses [11, 12], existing scholarly works have uncovered that station time distribution characteristics vary according to the surrounding land use [13], and different stations with unique passenger flow characteristics are influenced differently by these attributes [1]. While existing studies have explored the main features or partial segments of passenger flow distributions, the diversity in time distribution patterns at stations means that research based solely on extracted features cannot fully reveal the mechanisms behind these distributions. By dissecting the fundamental component curves of passenger flow distributions and studying the impact of land use on these curves, we can more effectively determine how different land uses influence overall daily passenger flow distributions, thereby better guiding the adjustment of passenger flow distributions through land use mix.

Understanding ridership distribution in metro stations from a land-use perspective is vital for optimizing urban transportation. Previous studies highlight the importance of infrastructure design and environmental factors. For instance, the optimization of infrastructure design [14, 15] provide valuable methodologies. Studies on airflow dynamics [16, 17] and the impact of air fences geometry [18] underscore the significance of environmental and structural integration in transportation planning. Building on these insights, our research explores how land uses affect the temporal distribution of metro ridership, aiming to enhance urban transportation systems and accommodate future growth.

Moreover, due to urban locational effects, the intensity of various factors' impact on station passenger flows exhibits spatial heterogeneity [19]. Although Ordinary Least Squares (OLS) is the most commonly used model for analyzing passenger flow characteristics and their influencing factors [20–22], it overlooks the spatial variations of their impact [23]. The Geographically Weighted Regression (GWR) model, which effectively accounts for spatial heterogeneity [24–26], has been applied to study station-level spatial distribution patterns, revealing relationships with land use [27], employment density [28], and station attributes [29].

In light of this, our research focuses on the Xi'an metro system, employing principal component analysis (PCA) to dissect the temporal distribution of station ridership and explore the main patterns of passenger flow distribution and their factor loadings across different stations. Building on this, we use the GWR model based on the surrounding built environment to analyze the spatial dependencies between the main temporal distribution patterns and land use. This study fundamentally reveals the mechanisms by which temporal distribution patterns are influenced by surrounding land use features, providing theoretical support for balancing commuter needs across different times of the day through mixed land use features, thus reducing the discrepancies between peak and off-peak travel volumes.

## Methodology

### Decomposition of the temporal distribution of metro ridership at the station level

The temporal distribution of station ridership varies and is influenced by multiple factors. To identify the fundamental patterns of passenger flow and determine the primary influencing

factors, this study employs PCA to decompose the time distribution of station ridership, extracting the key temporal distribution patterns. While we acknowledge that other dimensionality reduction techniques could also be considered, such as t-SNE or Factor Analysis, PCA was particularly suitable given its ability to handle the type of data we analyzed and its widespread use in similar research contexts [30, 31]. Future research could explore the application of alternative techniques to compare results and potentially uncover additional insights.

First, ridership at each station is standardized, as shown in Eq (1):

$$Z = \frac{Z' - \mu}{\sigma} \tag{1}$$

where:

$Z$ represents the result of the standardized processing.

$Z'$ represents the original passenger flow at each station.

$\mu$ represents the mean value of original passenger flow at various times throughout a single day for each station.

$\sigma$ represents the standard deviation of original passenger flow at various times throughout a single day for each station.

PCA effectively captures the majority of the information in the original variables by utilizing a small number of mutually independent components. The specific model is as follows:

Using the correlation coefficient matrix $\mathbf{L}$, the eigenvalues $\lambda_1 \geq \lambda_2 \geq \ldots \geq \lambda_q \geq 0$ and eigenvectors $\mathbf{I}(1), \mathbf{I}(2), \ldots, \mathbf{I}(q)$ are calculated, among them:

$$\mathbf{I}(i) = [l_1(i) \cdots l_q(i)]^T \in \mathbf{L}, \ i = 1, 2, \ldots, q \tag{2}$$

Then, we can obtain $q$ principal components as follows:

$$\mathbf{Y}_j(i) = \mathbf{I}(j)^T \mathbf{Z}(i), \ j = 1, 2, \ldots, q \tag{3}$$

where:

$\mathbf{Z}(i)$ is a $q$-dimensional random vector. $Yj(i)$ is the eigenvalue of the $j$th principal component.

The variance contribution rate of the $k$th principal component is given by:

$$\alpha_k = \lambda_k \bigg/ \sum_{i=1}^{q} \lambda_i \tag{4}$$

where:

$\lambda_j$ is the eigenvalue of the $j$th principal component $Yj(i)$.

Since it is desirable to reduce the original variables in practice, we choose the first $m(m<q)$ principal components to replace the original variables. The cumulative variance contribution rate of the principal components $Yj(i), \ldots, Ym(i)$ is given by:

$$\alpha = \sum_{j=1}^{m} \lambda_j \bigg/ \sum_{i=1}^{q} \lambda_i \tag{5}$$

Selecting a certain cumulative variance contribution rate $\alpha(0 \leq \alpha \leq 1)$, we can determine the number of principal components, denoted as $m$, as well as the variance contribution rate corresponding to each principal component.

By combining the standardized boardings and alightings for each time period, we construct a temporal distribution matrix of passenger flow, with rows representing stations and columns

representing time periods.

$$\mathbf{Z} = [\mathbf{Z}(1)^T \mathbf{Z}(2)^T \cdots \mathbf{Z}(n)^T] = \begin{bmatrix} Z_{11} & Z_{12} & \cdots & Z_{1n} \\ Z_{21} & Z_{22} & \cdots & Z_{2n} \\ \vdots & \vdots & \ddots & \vdots \\ Z_{q1} & Z_{q2} & \cdots & Z_{qn} \end{bmatrix} \tag{6}$$

where:

$q$ represents the number of stations;

$n$ represents the number of time intervals throughout the day.

PCA is employed to extract the top $m$ principal components, where $l_j(i)$ represents the factor loading of principal component $Yj(i)$ at the $i$th station, and the time distribution matrix $T_j$ of the principal component $Yj(i)$ can be represented as:

$$\mathbf{T}_j = \sum_{i=1}^{q} l_j(i)\mathbf{Z}(i) \Big/ q \tag{7}$$

## Geographical weighted regression (GWR) model

The GWR model, an extension of the OLS method, integrates spatial parameters to account for the spatial characteristics of the research subject. Unlike the Ordinary Least Squares (OLS) model, which assumes that the relationship between dependent and independent variables is constant across the study area, the GWR model acknowledges that these relationships can change from one location to another.

This local spatial analysis technique can be represented as follows:

$$y_i = \beta_{0(u_i,v_i)} + \sum_{k=1}^{p} \beta_{k(u_i,v_i)} x_{ik} + \varepsilon_i \tag{8}$$

where:

$y_i$ represents the factor loading of the $i$th station's principal component,

$x_{ik}$ represents the $k$th influencing factor of the $i$th station,

$(u_i,v_i)$ represents the coordinates of the $i$th station,

$\beta_{k(ui,vi)}$ represents the $k$th regression coefficient on the $i$th station, which is a function of spatial location,

$\beta_{0(ui,vi)}$ represents the constant term on the $i$th station,

$\varepsilon_i$ represents the error term of normal distribution for the $i$th station,

$p$ represents the number of influencing factors.

The expression for omitting the spatial position factor term is:

$$y_i = \beta_{i0} + \sum_{k=1}^{p} \beta_{ik} x_{ik} + \varepsilon_i \tag{9}$$

Formula (9) can be represented in matrix form as follows:

$$y_i = \mathbf{X}(i)\boldsymbol{\beta}(i) + \varepsilon_i \tag{10}$$

Using least square method, the regression parameters of point $i$ that minimize Eq (10) are:

$$\sum_{j=1}^{n} \omega_{ij} \cdot \left( y_i - \beta_{i0} - \sum_{k=1}^{p} \beta_{ik} x_{ik} \right)^2 \tag{11}$$

where:

$\omega_{ij}$ represents the spatial distance function between the variable of station $i$ and station $j$.

The spatial weight matrix influences the fitting results of the model [32]. In this study, the spatial weights are calculated using Gaussian distance:

$$\omega_{ij} = \exp\left[ -(\frac{d_{ij}}{b})^2 \right] \tag{12}$$

where:

$d_{ij}$ represents the spatial distance between station $i$ and station $j$.

$b$ represents the kernel bandwidth parameter.

The vector representation of the regression coefficients for station $i$ is as follows:

$$\hat{\boldsymbol{\beta}}(i) = [\boldsymbol{X}^T \boldsymbol{W}(i) \boldsymbol{X}]^{-1} \boldsymbol{X}^T \boldsymbol{W}(i) y \tag{13}$$

Subsequently, stepwise regression calculations are performed, resulting in the estimation matrix of regression parameters:

$$\boldsymbol{\beta} = \begin{pmatrix} \beta_{10} & \cdots & \beta_{n0} \\ \vdots & \ddots & \vdots \\ \beta_{1p} & \cdots & \beta_{np} \end{pmatrix} \tag{14}$$

Finally, the dependent variable is estimated according to the following formula:

$$\hat{\boldsymbol{y}}(i) = \boldsymbol{X}(i) \hat{\boldsymbol{\beta}}(i) \tag{15}$$

## Data collection

Xi'an, the capital of Shaanxi Province and the central city of the Guanzhong Plain Urban Agglomeration, inaugurated its first subway line in September 2011. By May 2021, Xi'an has developed a comprehensive subway network comprising 8 lines and 153 stations, as shown in Fig 1.

This study focuses on the Xi'an Metro system in 2021. The specific data used in this research includes:

① The metro card swiping data was supplied by the Xi'an Rail Transit Group Co., Ltd. This dataset includes processed data on boardings and alightings for 153 stations, covering the entire operational day from 4:00 AM to 00:00 AM, segmented into 30-minute intervals. We specifically used data from 19 working days in May 2021. The totaling data points is 232,560 (40 time intervals × 153 stations × 19 days × 2) accounting for both boardings and alightings. For our analysis, we calculated the average boardings and alightings for each 30-minute interval, from 4:00 AM to 00:00 AM, across these 19 working days at each station.

② The latitude and longitude coordinates of 153 stations obtained from Amap (a popular mapping service in China, https://www.amap.com).

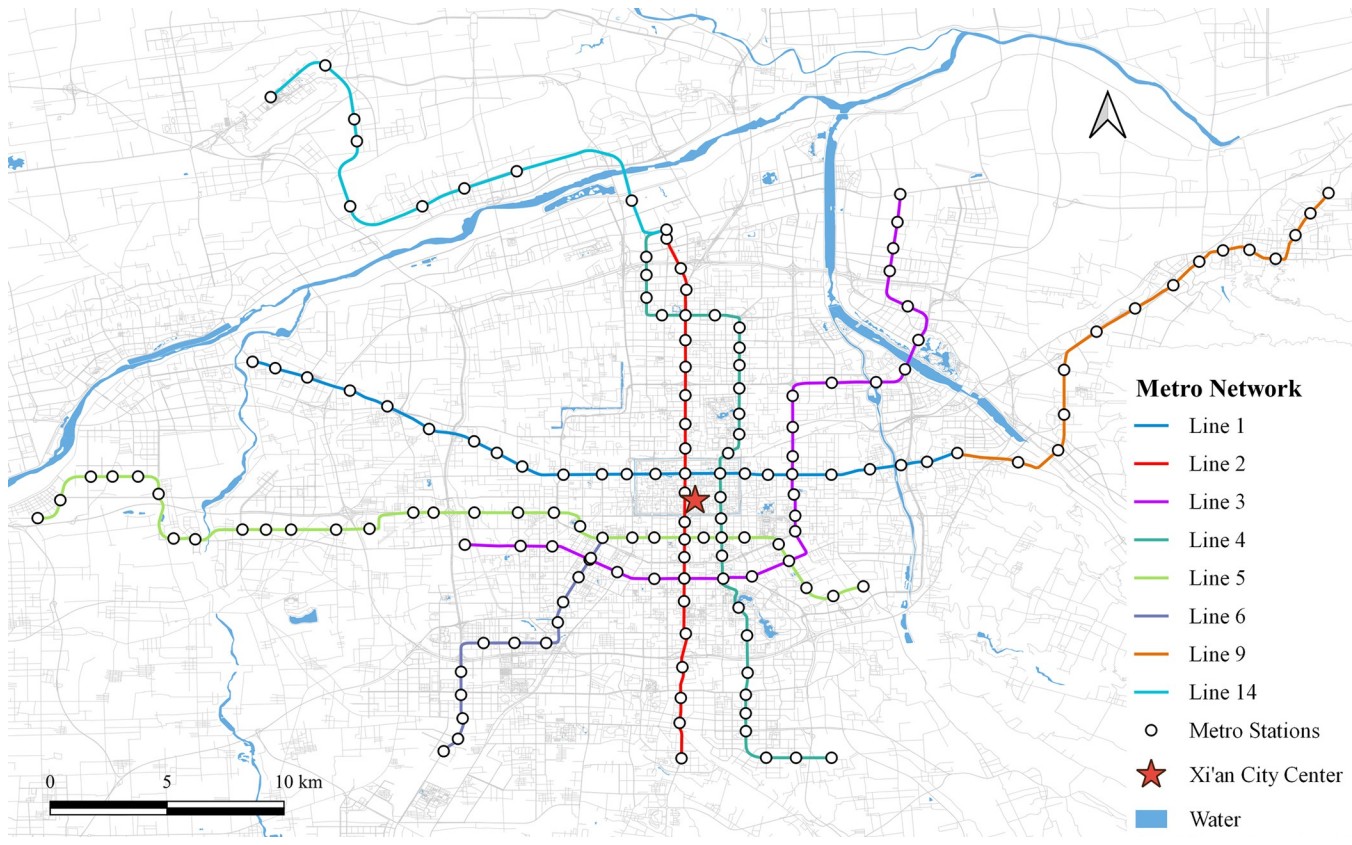

**Fig 1. The study area.** Note: self-drawn image. Source: Metro Stations and Metro Lines: Amap, Background: Open Street Map.

③ The land area within an 800-meter catchment area around each station was measured based on building outlines obtained from the Google earth (https://www.google.cn/intl/zh-en/earth/). The land attributes were then determined using points of interest (POI) data, obtained from the "search POI" interface within the web services of the Amap API open platform (https://lbs.amap.com/api/webservice/guide/api-advanced/search). Note that while there is no universally mandated distance threshold for stations' catchment areas, the 800-meter catchment area is widely recognized and supported in urban transportation research, particularly in studies examining the built environment's effects on metro ridership [33, 34].

## Model specification

In this study, we employed both OLS and GWR models to analyze how land use attributes influence the principal components of temporal distribution of metro ridership at different locations. Specifically, the GWR model enables the assignment of unique coefficients to each station, effectively capturing how the influence of land use attributes varies spatially. This allows us to observe local variations and identify areas where certain land uses have more or less impact on the temporal distribution of metro ridership.

The dependent and independent variables for this analysis are specified as follows:

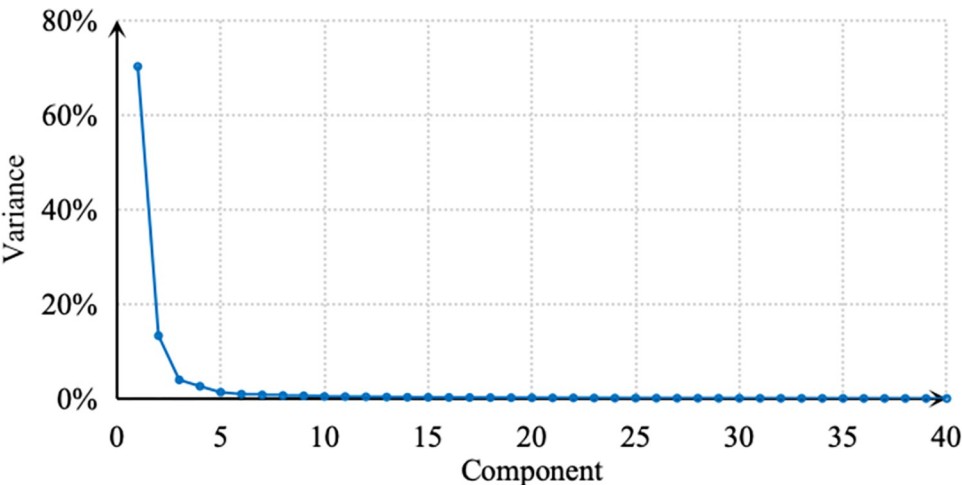

**Fig 2. Lithotripsy diagram of principal component proportion of passenger flow temporal distribution.** Note: self-drawn image.

## Dependent variables

According to Eq (1), the temporal distribution of ridership at each station is standardized. The average values of standardized boardings and alightings on weekdays are combined, with stations as columns (153 in total) and time intervals as rows (40 in total). This resulted in an 80x153 data matrix, where the first 40 rows represent daily boardings, and the last 40 rows represent daily alightings. PCA is performed on this data matrix. The principal components are ranked in descending order based on their contribution to the temporal distribution, as illustrated in Fig 2.

The first principal component accounts for 70.21% of the variance, representing the fundamental temporal distribution pattern at stations. The third principal component is located at the curve's inflection point, and the cumulative contribution rate of the first three principal components is 87.48%. Therefore, the subsequent analysis focuses on the temporal distribution of station ridership using the first three principal components as the dependent variables.

Based on Eq (7), the temporal distribution derived from the first three principal components can be calculated separately, as shown in Fig 3.

Specifically, the first principal component, as shown in Fig 3(A), exhibits a bimodal pattern resembling a "saddle shape" in both boardings and alightings. The peak times for the boarding curve occur between 8:00–8:30 AM and 6:00–6:30 PM, while the peaks for the alighting curve are delayed by half an hour. These peaks coincide with the city's morning and evening peak hours, reflecting the characteristics of inelastic travel patterns that are typical of commuter traffic.

The peak hours of the second principal component (see Fig 3(B)) correspond to those of the first principal component and similarly coincide with the city's morning and evening peak hours, but exhibiting positive and negative variations. This pattern implicitly reflects the primary function of this principal component, that is to adjust the shape of the temporal distribution during peak hours.

The distribution curve of the third principal component, illustrated in Fig 3(C), reveals negative peaks for both boardings and alightings during the morning peak period (8:00–8:30 AM). Additionally, boardings during the evening peak (6:00–6:30 PM) are also marked by negative values, while positive values are observed during other non-peak periods. This pattern suggests that the third principal component functions primarily to balance the disparities between peak and off-peak periods through its factor loadings.

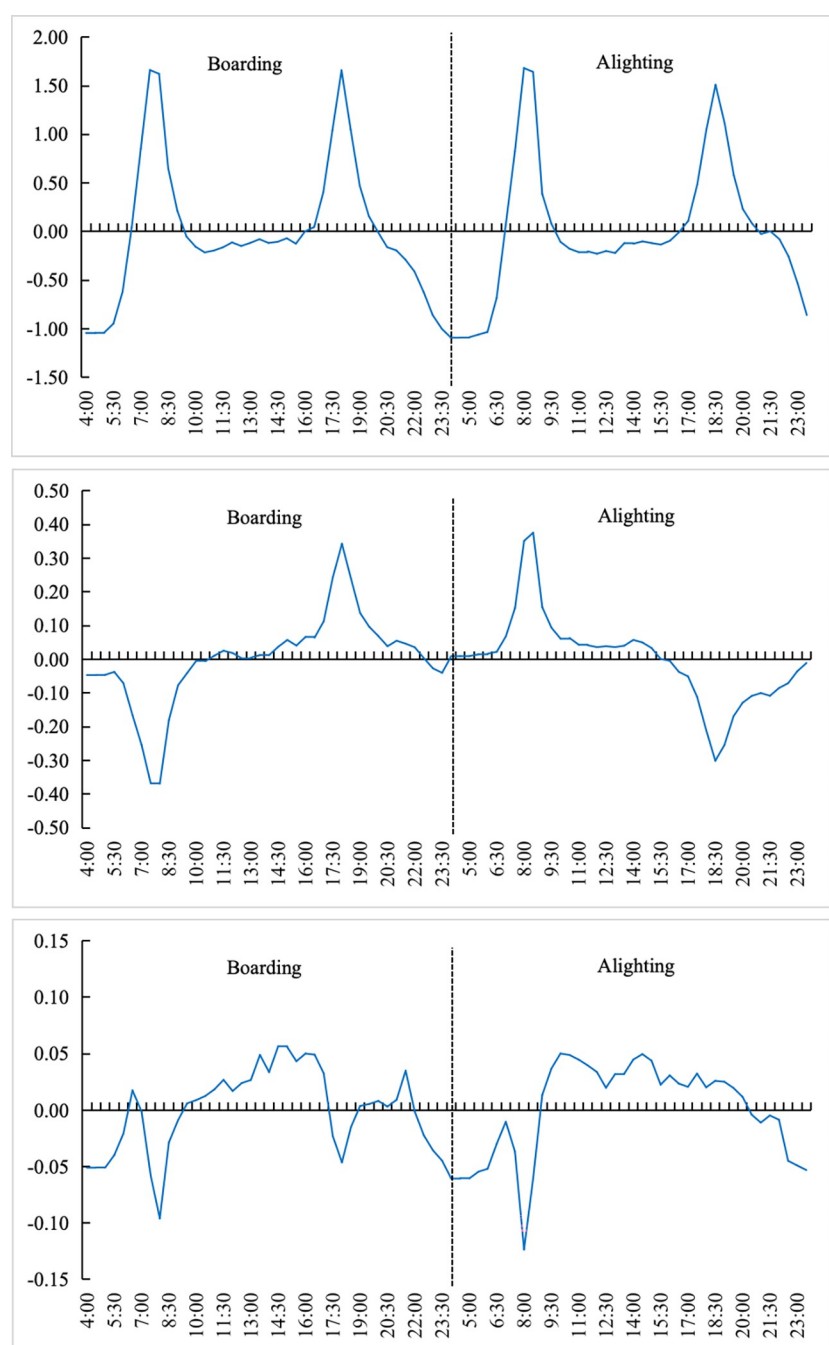

**Fig 3.** Component 1 to component 3 of decomposition of passenger flow temporal distribution (a) (b) (c). Note: self-drawn image.

Therefore, the empirical interpretations of the three principal components of the temporal distribution of metro ridership are summarized as follows:

- **The first principal component** influences the peak hour ridership, reflecting the characteristics of mandatory travel patterns. This component captures the core usage patterns associated with commuter travel during rush hours.

- **The second principal component** adjusts the shape of the temporal distribution during peak hours. This modification involves refining the intensity, duration, and direction of peak period ridership, possibly addressing fluctuations within those peak times.

- **The third principal component** aims to balance the disparities between peak and off-peak periods. It functions to smooth out the distribution of ridership across the day, reducing the extremity of rush hour peaks while enhancing off-peak ridership levels.

Analysis of the principal components reveals that the fundamental temporal pattern of metro station passenger flow is primarily driven by inelastic travel demand. This finding is not surprising but remains crucial for transit planners, who must pay attention to the critical periods when metro stations are under the most pressure. This is essential for designing station capacity and configuring facilities and equipment to handle the highest demand effectively. Land use that induces commute travel near subway stations should also be carefully considered, and such development should be limited for stations that already experience severe peak-off-peak imbalance issues.

Additionally, it is important for the metro system to cater to non-inelastic travel demands on weekdays as well to ensure sustained passenger traffic during non-peak periods. Transit planners should also be aware of scenarios where off-peak ridership may exceed traditional peak periods at specific stations, potentially influenced by leisure land uses, leading to a phenomenon known as peak deviation. This insight should be incorporated into ridership forecasting and metro network planning to enhance system efficiency and user satisfaction [35].

## Independent variables

Given the close relationship between metro ridership and nearby land use features, this study selects building area of each land use type as an independent variable. Based on the land use classification standards from the Landsearch website (https://www.landsearch.com/blog/land-use-types-definitions) it is categorized into eight types, as presented in Table 1. Multiple collinearity and spatial autocorrelation tests on these lands are conducted, with the results also summarized in Table 1. The Variance Inflation Factor (VIF) values for all variables are found to be less than 5, indicating no significant collinearity issues among the independent variables [36]. The Moran's I values are positive for all variables, suggesting significant spatial autocorrelation among the variables. These results satisfy the conditions necessary for establishing the GWR model.

**Table 1. Test results of multicollinearity and spatial autocorrelation.**

| Independent variable | Min value | Max value | VIF | Moran'I | Z-value |
|---|---|---|---|---|---|
| Residential *** | 0 | 5.272 | 2.888 | 0.370 | 9.179 |
| Educational *** | 0 | 1.041 | 1.602 | 0.568 | 14.314 |
| Commercial & office *** | 0 | 2.727 | 2.194 | 0.513 | 12.640 |
| Cultural & entertainment | 0 | 0.622 | 1.190 | 0.046 | 1.452 |
| Healthcare *** | 0 | 0.561 | 1.289 | 0.305 | 7.849 |
| External hub *** | 0 | 1.377 | 1.912 | 0.060 | 3.045 |
| Recreational & park *** | 0 | 1.074 | 1.933 | 0.099 | 2.612 |
| Industrial *** | 0 | 1.457 | 3.246 | 0.375 | 9.396 |

*** p-value<0.001

** p-value<0.01

* p-value<0.05, · p-value <0.1

**Table 2. Comparative analysis of model results.**

| Dependent variable | Model | BIC/MDL | AICc | Adjusted $R^2$ |
|---|---|---|---|---|
| Principal component 1 | OLS | -204.899 | -225.391 | 0.625 |
| | GWR | -157.764 | -237.363 | 0.712 |
| Principal component 2 | OLS | 99.683 | 82.076 | 0.283 |
| | GWR | 144.935 | 58.987 | 0.509 |
| Principal component 3 | OLS | -49.276 | -69.717 | 0.111 |
| | GWR | -52.764 | -135.084 | 0.525 |

## Results

### Geographically weighted regression results

In this study, the same land use variables are selected to explain the three principal components. After normalizing these variables, both OLS and GWR models are established for comparative analysis, with the results shown in Table 2.

It is observed that for all three principal components, the GWR models outperform the OLS models in terms of smaller AICc values and greater adjusted $R^2$ values. This indicates that the GWR models, which consider spatial factors, provide better regression results compared to the OLS models. This highlights the implicit importance of accounting for spatial heterogeneity in urban transit studies.

Comparatively, lower Adjusted $R^2$ values are observed for the second and third principal components in both the OLS and GWR models. This is understandable, as the second component is designed to adjust for peak traffic during morning and evening rush hours, reflecting temporal variations in ridership. The third captures the stochastic nature of passenger flow fluctuations, both of which are less predictable and more random.

The regression coefficients for the GWR models are presented in Table 3. Note that, in these models, a positive coefficient associated with the first principal component signifies an increase in commute-related travel, reinforcing the typical morning and evening peak hours. For the second principal component, a positive coefficient suggests a decrease in morning peak boardings and evening peak alightings, while increasing morning peak alightings and evening peak boardings. This pattern intensifies the tidal characteristics typical of workplace-dominated stations. A negative coefficient represents the opposite effect on the temporal distribution, exacerbating the tidal features of home-dominated stations. For the third principal component, a positive coefficient act to smooth the distribution curve by reducing peak passenger flows while increasing off-peak flows, whereas a negative coefficient accentuates the peaks and troughs, making the distribution curve sharper.

**Table 3. Regression coefficients of GWR model.**

| Independent variable | Principal component 1 | Principal component 2 | Principal component 3 |
|---|---|---|---|
| Residential land | 0.035 | -0.15 | -0.008 |
| Educational land | 0.147 | --- | --- |
| Commercial & office land | --- | 0.085 | 0.005 |
| Healthcare land | -0.075 | 0.016 | --- |
| External hub land | 1.251 | --- | 0.132 |
| Recreational & park land | -0.022 | --- | 0.006 |
| Industrial land | --- | -0.101 | --- |
| Constant term | 0.962 | -0.065 | -0.009 |

The results show that for the first principal component, residential, educational and external hub land uses have positive coefficients, indicating that these land use types increase commuting activity at the stations. On the other hand, healthcare and recreational & park land uses tend to suppress the morning and evening peak values of the ridership temporal distribution. This suggests that while residential, educational, and hub areas drive peak commuting times, healthcare and recreational areas help spread ridership more evenly throughout the day, reducing peak congestion.

In the results for the second principal component, the coefficients for residential and industrial land uses are negative. Comparing the shape of its curve, it is evident that these land uses function as sources of passenger flow during the morning peak and as attractors during the evening peak. In contrast, commercial & office and healthcare lands have positive coefficients, suggesting they act as attractors for morning peak flow and as sources for evening peak flow. This pattern highlights the distinct roles different land use types play in influencing the temporal distribution of metro ridership. Stations serving these particular single land use functions exhibit tidal temporal patterns; for example, residential and industrial-dominated stations are busy with outbound morning commutes and inbound evening returns.

For the third principal component, only the coefficient for residential land use is negative. When combined with its coefficient for the first principal component, it becomes clear that residential land use not only increases the share of peak passenger flow but also reduces the proportion of off-peak passenger flow. Conversely, commercial, office, external hub, and recreational & park land uses help to alleviate the concentration of peak passenger flows and promote a more balanced temporal distribution of passenger flow. This finding underscores the role of diverse land uses in achieving a smoother ridership pattern, highlighting the importance of mixed-use development in reducing peak load strain on metro systems.

The combined effects of various land uses on the three principal components reveal that residential and educational land uses are primary factors in shaping the morning and evening peak flows at metro stations. This is expected as these land use types are the main sources of mandatory trips, playing a critical role in the utilization efficiency of the metro system. While these trips make the peaks sharper, they still signify a stable source of metro usage.

Conversely, commercial & office, healthcare and recreational & park land uses show importance in balancing the temporal distribution of passenger flows at metro stations. These types of land use effectively soften the sharpness of passenger flows during peak hours by redistributing some of the peak hour flows to off-peak times. Specifically, the integration of commercial-office complexes in modern urban development not only provides office spaces but also enhances urban vibrancy. From the perspective of metro system, such developments boost commercial activities during off-peak periods, thereby alleviating the pressure of ridership during peak hours. Healthcare and recreational & park land uses attract passenger flows with more dispersed time distributions, further reducing the concentration of passengers during peak hours.

By strategically allocating and managing these land use types, it is possible to significantly mitigate the peak-off-peak differences, thus enhancing metro operational efficiency and service quality.

External hub makes significant contributions to both peak and off-peak passenger flows at metro stations. By providing a large number of employment opportunities, these hubs significantly promote the formation of ridership during the morning and evening peak hours. Additionally, as critical nodes connecting the city with external areas, external hubs ensure a continuous and diverse flow of passengers, leading to a more balanced temporal distribution of passenger flows across the metro system. However, the location selection of external hubs for a metropolitan area is influenced by many factors and is difficult to modify during the

normal urban land use planning process. Therefore, this aspect is not further discussed here in the context of achieving peak and off-peak balance.

## The spatial effect of variables on the temporal distribution patterns at stations

To further investigate the impact of local variables on different metro stations, this study primarily focuses on the spatial effects of residential, educational, and commercial & office land uses. On one hand, these land uses directly influence the demand for metro services by generating significant passenger flows during peak times. Understanding their impacts is crucial for managing peak loads. On the other hand, these land uses are often the focus of urban planning and policy-making efforts aimed at transit-oriented development, so analyzing their effects can provide insights that are directly applicable to planning decisions, helping to enhance public transit efficiency and urban livability.

Regarding residential land use, its impact on the first principal component varies spatially within the range of [0.005, 0.087], as shown in Fig 4. While, the overall influence is relatively low and varies slightly, it is consistently positive, indicating that residential land use positively affects peak-hour passenger flow during weekdays. The most significant impact is observed in the northern areas beyond the river and at the western terminal stations of the metro network. These periphery regions have slower development, leading most residents to commute to

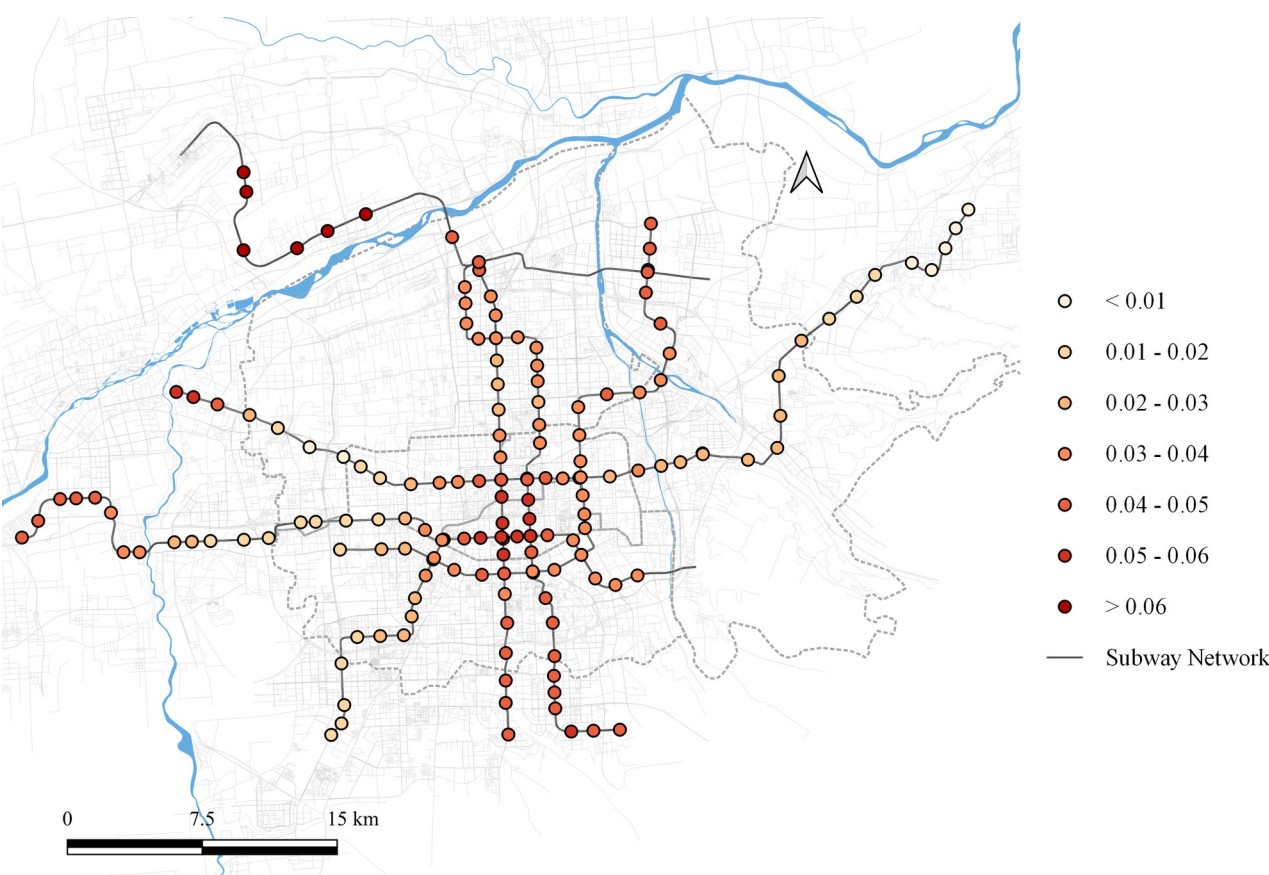

**Fig 4. Local parameter distribution of residential land to component 1.** Note: self-drawn image. Source: Metro Stations and Metro Lines: Amap, Background: Open Street Map.

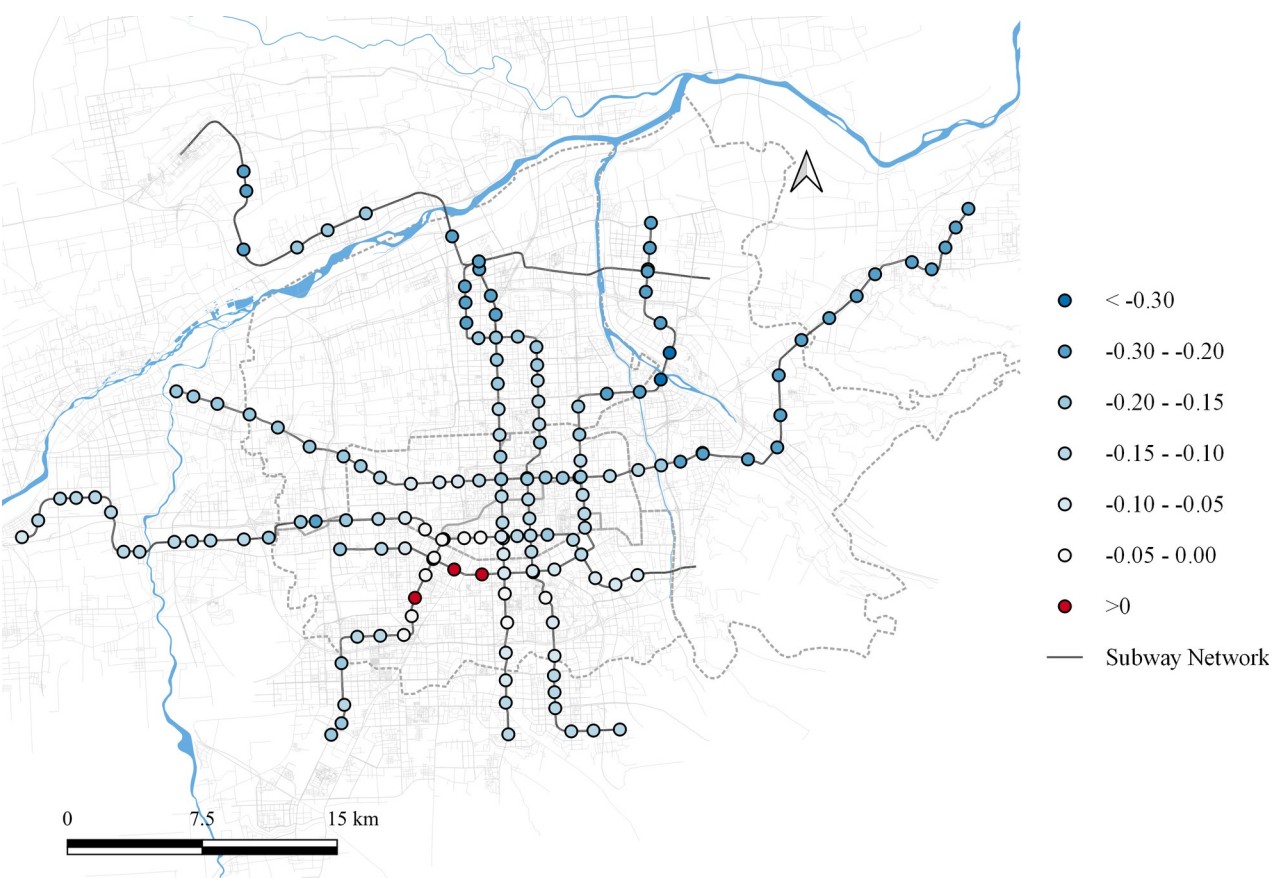

**Fig 5. Local parameter distribution of residential land to component 2.** Note: self-drawn image. Source: Metro Stations and Metro Lines: Amap, Background: Open Street Map.

other parts of the city for work, thereby contributing significantly to peak-hour passenger flows. In the city center, the impact of residential land is also notable. This is attributed to Xi'an's policy of halting new residential developments in the central area to protect the ancient Ming City Wall. As the city center increasingly becomes a commercial hub, residents in these areas also need to commute to other regions for employment, resulting in a significant influence of residential land use on commuting flows.

The impact of residential land use on the second principal component ranges from [-0.318, 0.039], as shown in Fig 5. In most areas, residential land use negatively influences the factor loadings of the second principal component. However, a slight positive effect is observed at some stations in the southwestern part of the city center. This may be attributed to the presence of numerous private enterprises in the area, resulting in residential land use that also contain office spaces. This blend gives the area attributes of both a source and an attractor during the morning and evening peaks.

The impact of residential land use on the third principal component ranges from [-0.058, 0.047], as shown in Fig 6. The overall spatial distribution pattern is similar to that of the first principal component. In areas where residential land use has a significant influence on the first principal component, it suppresses the third principal component, resulting in sharper morning and evening peaks in passenger flow. Conversely, in areas with minimal influence on the first principal component, residential land use positively affects the third principal component, leading to a smoother passenger flow time distribution curve.

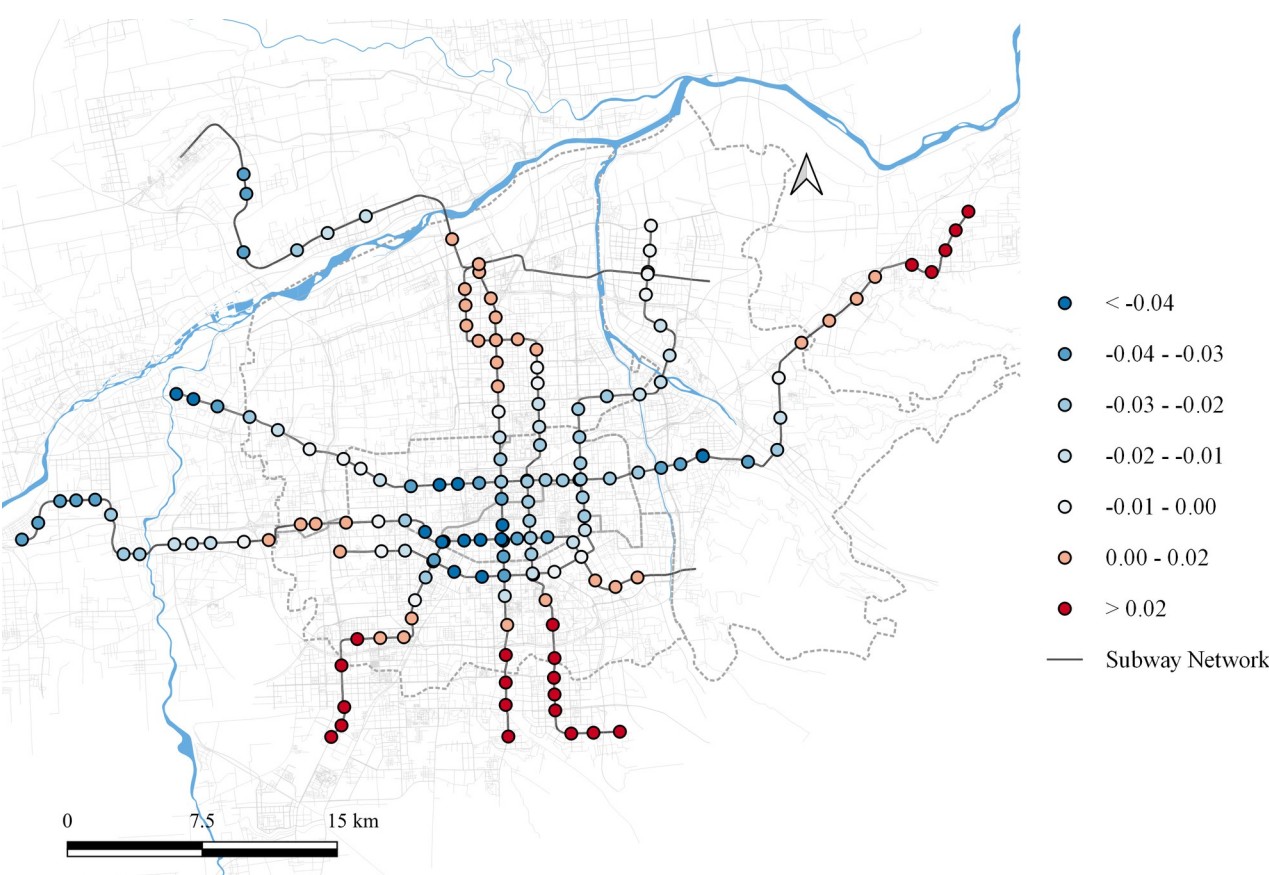

**Fig 6. Local parameter distribution of residential land to component 3.** Note: self-drawn image. Source: Metro Stations and Metro Lines: Amap, Background: Open Street Map.

Educational land use only shows significant impacts on the first principal component, with an influence range between [-0.145, 0.526], as shown in Fig 7. This influence can be broadly categorized into three parts: the central-southern area shows almost no impact; the northern area exhibits a substantial positive impact, intensifying the passenger flows in the morning and evening peaks; and the eastern and western areas show a moderate negative impact, diminishing the peak passenger flows.

In China, the policy of attending schools within one's residential district means students typically attend schools within their local area. The southern region of Xi'an serves as the city's educational and cultural center, housing numerous primary and secondary schools, universities, and research institutes. Therefore, most residents in this area do not need to use metro for school. In contrast, the northern region has relatively fewer educational resources, so the longer distances are required for school travel, leading to increased passenger flow during the morning and evening peaks on the metro network.

Commercial & office land use impacts both the second and third principal components. For the second principal component, the impact range is between [-0.272, 0.271], as shown in Fig 8. In the southern region, commercial & office land use highlights its role in generating peak commute demand. This area is an emerging industrial zone in Xi'an, where high housing prices force many commuters to live in other areas, resulting in significant peak-hour passenger flows. Conversely, commercial & office land use at the western terminal stations of the

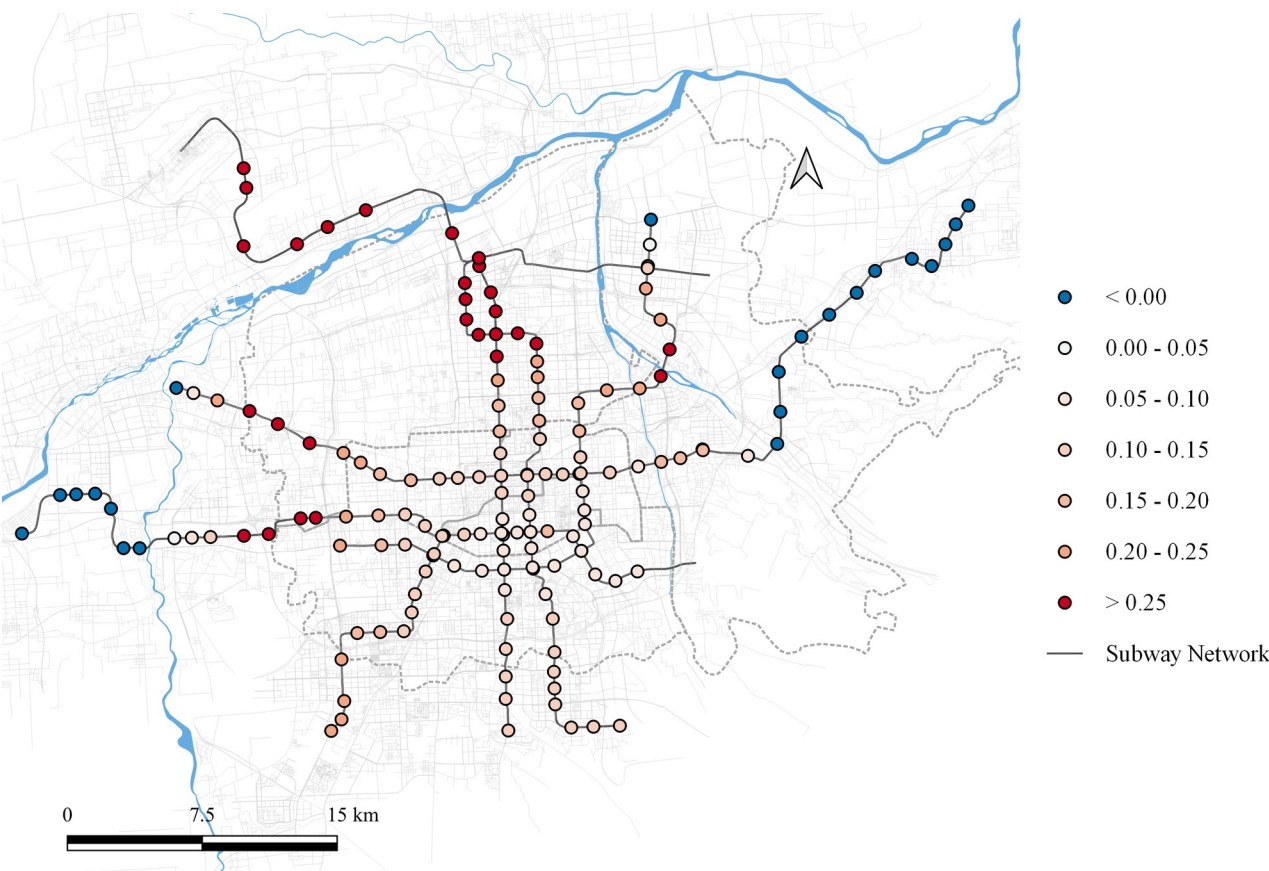

**Fig 7. Local parameter distribution of education land to component 1.** Note: self-drawn image. Source: Metro Stations and Metro Lines: Amap, Background: Open Street Map.

metro network negatively impacts commuter flow. Due to its immature development, the influence of commercial & office land in this area is weak, and it fails to act as a significant attractor of passenger flow.

Its impact on the third principal component ranges between [-0.105, 0.069], as shown in Fig 9. In the central and southern regions, commercial & office land use exerts a positive influence, while in the eastern, western, and northern regions, it has a negative influence.

As discussed earlier, a positive influence of the third principal component leads to a smoother passenger flow distribution, while a negative influence sharpens the temporal distribution curve. Accordingly, this spatial impact indicates that the central and southern regions, being the economic centers of Xi'an, showcase the commercial attributes of the commercial & office land use. While, in the eastern, western, and northern regions, the commercial & office land use emphasizes its office attributes due to the lower commercial impact in these areas.

## Conclusion

### Research findings

This paper employs PCA to decompose the temporal distributions of passenger flow at metro stations in Xi'an, and identify their main patterns and factor loadings. A GWR model is then applied to analyze the spatial dependency relationship between these factor loadings and the surrounding land use attributes.

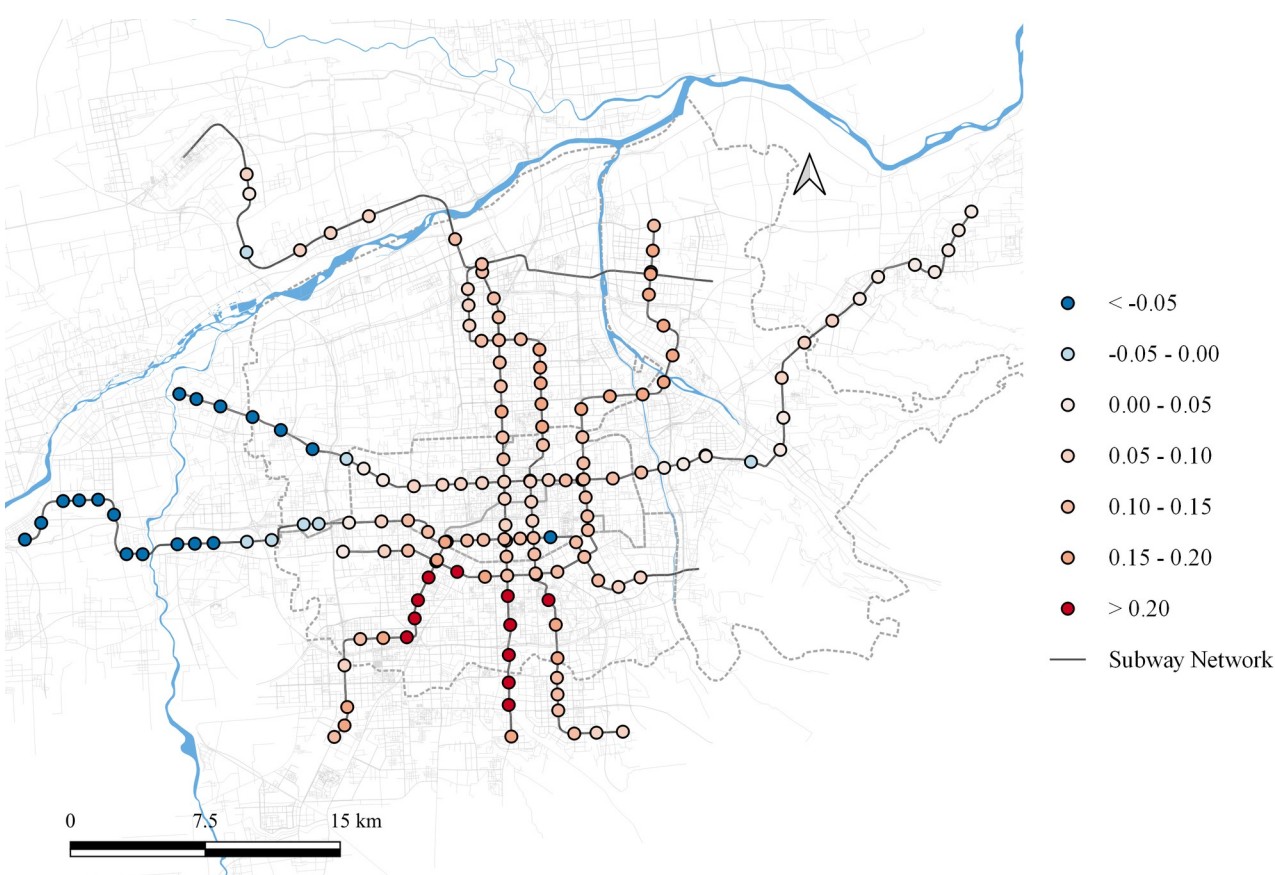

**Fig 8. Local parameter distribution of commerce and office land to component 2.** Note: self-drawn image. Source: Metro Stations and Metro Lines: Amap, Background: Open Street Map.

The results from PCA reveal that subway stations exhibit diverse temporal distributions of passenger flows, with several predominant temporal distribution patterns varying in proportion across different stations. Wherein, the first principal component, which dominates on weekdays, displays clear characteristics of inelastic travel distributions. The second principal component also matches these peak hours but varies positively and negatively, effectively inverting the peak values of passenger flows at stations and shifting the distribution towards a unimodal shape. The third principal component smooths the time distribution curve by reducing peak-hour flows and enhancing off-peak flows.

Different types of land use affect these principal components in varying ways. Residential and educational land uses are major drivers of the first principal component, facilitating the formation of morning and evening peak flows. Commercial & office, healthcare land uses have a significant positive impact on the second principal component, promoting a unimodal peak in passenger traffic. Meanwhile, recreational & park land use helps suppress peak flows and increase off-peak flows; whereas external hub land enhances passenger flows throughout the day.

The impact of different land uses varies spatially across the metro network. Residential land use, particularly in city centers and peripheral areas, has a pronounced effect on increasing commuter flows. Areas with a lower proportion of educational land use exert a greater impact on peak-hour flows at metro stations, likely due to factors such as increased travel distances

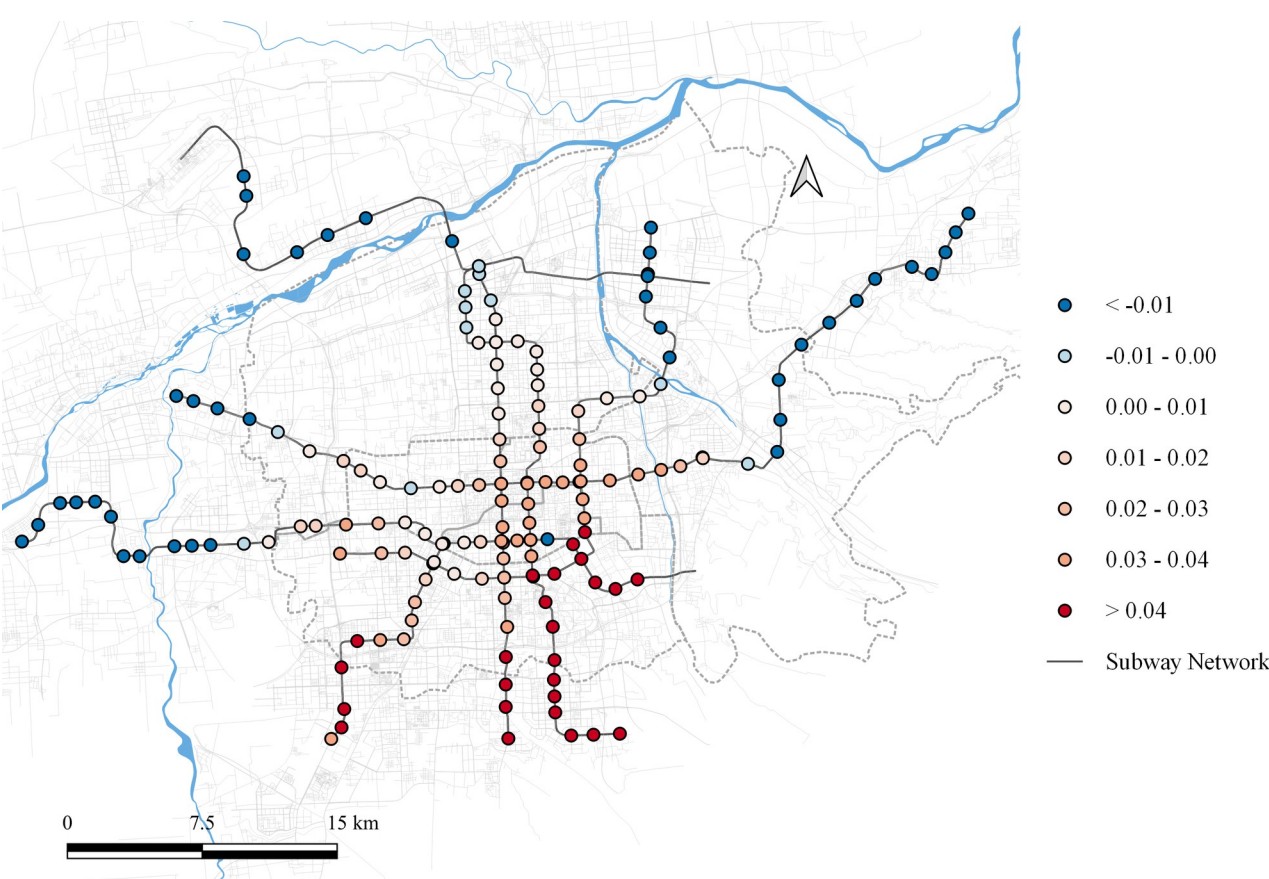

**Fig 9. Local parameter distribution of commerce and office land to component 3.** Note: self-drawn image. Source: Metro Stations and Metro Lines: Amap, Background: Open Street Map.

required for schooling. Commercial & office land use in urban office districts tends to stimulate the formation of morning and evening peak flows. Conversely, in economically active regions, commercial & office land use contributes to a smoother time distribution curve of passenger flows at stations, helping to distribute transit usage more evenly throughout the day.

## Implications and limitations

The Chinese government has endorsed a policy that encourages integrating urban rail transit projects with land use development along transit lines, fostering construction around station areas. Different types of land use variably affect the temporal distributions of stations ridership, which are key in shaping peak and off-peak hours. An important focus should be on balancing ridership from the perspective of land use development and adjustment, which first and foremost, can enable the metro system to meet demand effectively without the need for further expansion. Secondly, it helps minimize resource wastage during off-peak hours, ensuring that the transit system operates efficiently across all times of the day.

The results from this study suggest that adjusting the mix of land uses around stations, by effectively combining residential, commercial, office, educational, and recreational land uses, can "flatten the peaks and fill the valleys," promoting balanced utilization of the subway system throughout the day.

Land use and transportation strategies can also be tailored to the characteristics of different urban areas, such as city centers and outlying districts. For example, in city centers where passenger volumes are already high, the focus should be achieving the rational mix of land uses to balance daily passenger flows. In contrast, in peripheral areas where passenger volumes are lower, initial development should prioritize residential, commercial, and educational land use to foster essential commuting and schooling travel, thus enhancing passenger flows and regional travel dynamics. Subsequent developments can gradually include other types of land uses to balance passenger flows throughout the day.

These policy recommendations aim to optimize urban development and transportation planning in a way that aligns with evolving urban dynamics and improves the overall efficiency and sustainability of public transit systems.

However, this study is constrained by the granularity of the research data. Specifically, the available data only captures aggregated 30-min passenger flows, which restricts our ability to examine the formation of extremely high peak traffic conditions and their relationship with land use patterns. A more in-depth analysis could be expected in the future if original Automated Fare Collection swipe card data is made available, offering finer temporal resolution.

## Supporting information

**S1 File. Data used for modeling, including both dependent and independent variables.** (XLSX)

## Author Contributions

**Data curation:** Liqiang Yu.

**Formal analysis:** Lang Song.

**Funding acquisition:** Shian Dai.

**Investigation:** Liqiang Yu.

**Methodology:** Shian Dai, Lang Song.

**Software:** Shian Dai, Liqiang Yu.

**Supervision:** Ying Li.

**Validation:** Xuze Fan.

**Visualization:** Liqiang Yu.

**Writing – original draft:** Shian Dai.

**Writing – review & editing:** Ying Li.

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
