## [Decision Letter · Decision Letter 0]

31 May 2024

PONE-D-24-04620The Temporal Distribution of Ridership in Metro Stations from Land-use PerspectivePLOS ONE

Dear Dr. SONG,

Thank you for submitting your manuscript to PLOS ONE. After careful consideration, we feel that it has merit but does not fully meet PLOS ONE’s publication criteria as it currently stands. Therefore, we invite you to submit a revised version of the manuscript that addresses the points raised during the review process.

We look forward to receiving your revised manuscript.

Kind regards,

Dr. Walid Al-Shaar

Academic Editor

PLOS ONE

Journal Requirements:

3. Thank you for stating the following financial disclosure: "Youth Projects of Xi'an Jiaotong University City College [grant number: 2022Q01]"  

4. We note that Figures 1, 4, 5, 6, 7, 8, and 9 in your submission contain [map/satellite] images which may be copyrighted. All PLOS content is published under the Creative Commons Attribution License (CC BY 4.0), which means that the manuscript, images, and Supporting Information files will be freely available online, and any third party is permitted to access, download, copy, distribute, and use these materials in any way, even commercially, with proper attribution. For these reasons, we cannot publish previously copyrighted maps or satellite images created using proprietary data, such as Google software (Google Maps, Street View, and Earth). For more information, see our copyright guidelines: http://journals.plos.org/plosone/s/licenses-and-copyright.

a. You may seek permission from the original copyright holder of Figures 1, 4, 5, 6, 7, 8, and 9  to publish the content specifically under the CC BY 4.0 license.  

Reviewers' comments:

Reviewer's Responses to Questions

**Comments to the Author**

1. Is the manuscript technically sound, and do the data support the conclusions?

Reviewer #1: Yes

Reviewer #2: Partly

Reviewer #3: Yes

2. Has the statistical analysis been performed appropriately and rigorously? 

Reviewer #1: Yes

Reviewer #2: No

Reviewer #3: Yes

3. Have the authors made all data underlying the findings in their manuscript fully available?

Reviewer #1: Yes

Reviewer #2: Yes

Reviewer #3: Yes

4. Is the manuscript presented in an intelligible fashion and written in standard English?

Reviewer #1: Yes

Reviewer #2: No

Reviewer #3: Yes

5. Review Comments to the Author

Reviewer #1: 1. the aim need to specify in the abstract

2. line 207, the author used 800 meters, why use this value

3. need to clear the time period of data collection of 30 minutes intervals

4. Add ref. for land search website

5. R2 is low for the second and third components, need to explain.

6. line 301 need to justify the writing

7. add paragraph for the work limitations and recommendation.

Reviewer #2: Thanks for presenting the manuscript entitled "The Temporal Distribution of Ridership in Metro Stations from Land-use Perspective". In this study, the relationship between the land use ratio around urban rail transit stations and the temporal distribution of passenger flows is explored using methods such as principal component analysis and geographical weighted regression (GWR) models. This study provides an important reference for optimizing land use around metro stations. However, the manuscript fails to meet the publication criteria of PLOS ONE in the following key aspects:

1) The innovativeness and research significance are not clearly articulated. Further optimization and refinement of the thesis structure and writing presentation are required.

2) The experiments are too simple, and the comparative analyses are not sufficient to prove the reliability and sophistication of the experiments.

3) Abbreviations should be given in parentheses the first time a technical term, such as PCA is used, in the manuscript the font is too small in part of line 301, the initial letter in line 305 is not capitalized, and there are grammatical errors in parts of the article.

4) Figures 1 to 9 in this manuscript are not clear. Please insert the original figures.

Overall, I cannot recommend it to be published at the current stage. Hope my comments will help authors to improve the manuscript.

Reviewer #3: Subject: Review of Manuscript: “Temporal Distribution of Ridership in Metro Stations from a Land-use Perspective”

Dear Editor and Authors,

I have carefully reviewed the manuscript titled “Temporal Distribution of Ridership in Metro Stations from a Land-use Perspective.” Overall, the study provides valuable insights into the relationship between land use and passenger flow at urban rail transit stations.

I have identified several major comments and questions that I believe need to be addressed before the manuscript can be considered for publication. However, I recommend addressing the following points:

1. Data Collection and Representativeness:

o Could you provide more details about the data sources used for passenger flow analysis? How was the data collected, and what is the sample size?

o How representative is the dataset of the overall metro system ridership?

2. Introduction:

o The introduction provides a clear and concise overview of the research problem. I suggested the following articles added in the manuscript:

o "The analysis of utilizing multiple fences in high-speed tracks on the aerodynamic characteristics of a high-speed train model," Iranian Journal of Science and Technology, Transactions of Mechanical Engineering, 2023.

o "The Influence of Inclined Barriers on Airflow over a High Speed Train under Crosswind Condition, In A. G. Hessami and R. Muttram, Book: New Research on Railway Engineering and Transportation, IntechOpen, 2023, ISBN 978-1-83768-620-9.

o "The optimum model determination of porous barriers in high-speed tracks," Journal of Rail and Rapid Transit, Vol. 236, Issue 1, pp. 15-25, 2022.

o "Multi Objective Optimization of Aerodynamic Design of High Speed Railway Windbreaks using Lattice Boltzmann Method and Wind Tunnel Test Results", International Journal of Rail Transportation, 2018.

o "2D and 3D numerical and experimental analyses of the aerodynamic effects of air fences on a high-speed train", Journal of Wind and Structures, Vol. 32, No. 6, pp.539-550, 2021.

o "Two-dimensional analysis of the influence of windbreaks on airflow over a high-speed train under crosswind by using Lattice Boltzmann Method ", Journal of Rail and Rapid Transit, Vol. 232, Issue. 3, pp. 863-872, 2018.

o "The Impact of Air Fences Geometry on Air Flow around an ICE3 High Speed Train on a Double Line Railway Track with exposure to Crosswinds", Journal of Applied Fluid Mechanics, Vol. 11, No. 3, pp. 743-754, 2018.

3. Interpretation of Principal Components:

o In the PCA analysis, how were the principal components interpreted? Specifically, what practical implications do the first, second, and third components have for transit planning?

o Were any other dimensionality reduction techniques considered besides PCA?

4. Spatial Variation and GWR:

o Could you elaborate on the spatial dependence analysis using the geographically weighted regression (GWR) model? How does it account for spatial heterogeneity?

o Were there specific areas around Xi’an where the relationships between factor loadings and influencing factors differed significantly?

5. Land Use Factors and Recommendations:

o Which specific land use types (e.g., residential, commercial, healthcare) contribute most significantly to morning and evening peak flows?

o Based on your findings, what actionable recommendations can be made for optimizing land use around metro stations?

6. Practical Implications:

o Discuss how transit agencies and policymakers can apply the study’s findings to enhance metro system performance.

o Consider proposing specific interventions based on the research outcomes.

Thank you for your valuable contribution to the field. I look forward to seeing the revised manuscript.

Sincerely,

6. PLOS authors have the option to publish the peer review history of their article (what does this mean?). If published, this will include your full peer review and any attached files.

Reviewer #1: No

Reviewer #2: No

Reviewer #3: No

---

## [Author Response · Author response to Decision Letter 0]

12 Jul 2024

Please find our response to reviewers in the attached response letter.

---

## [Decision Letter · Decision Letter 1]

30 Jul 2024

The Temporal Distribution of Ridership in Metro Stations from Land-use Perspective

PONE-D-24-04620R1

Dear Dr. DAI,

We’re pleased to inform you that your manuscript has been judged scientifically suitable for publication and will be formally accepted for publication once it meets all outstanding technical requirements.

Kind regards,

Dr. Walid Al-Shaar

Academic Editor

PLOS ONE 

---

## [Editor Report · Acceptance letter]

22 Aug 2024

PONE-D-24-04620R1 

PLOS ONE

Dear Dr. DAI, 

I'm pleased to inform you that your manuscript has been deemed suitable for publication in PLOS ONE. Congratulations! Your manuscript is now being handed over to our production team.

Kind regards, 

on behalf of

Dr. Walid Al-Shaar 

Academic Editor

PLOS ONE